# Tool Use by Four Species of Indo-Pacific Sea Urchins

**Glyn A. Barrett [1,2,\*]**, **Dominic Revell [2]**, **Lucy Harding [2]**, **Ian Mills [2]**, **Axelle Jorcin [2]** **and Klaus M. Stiefel [2,3,4]**

1 School of Biological Sciences, University of Reading, Reading RG6 6UR, UK
2 People and The Sea, Logon, Daanbantayan, Cebu 6000, Philippines; domrevell@gmail.com (D.R.);
  lucy@peopleandthesea (L.H.); Ian@peopleandthesea.org (I.M.); axelle@peopleandthesea.org (A.J.);
  klaus@neurolinx.org (K.M.S.)
3 Neurolinx Research Institute, La Jolla, CA 92039, USA
4 Marine Science Institute, University of the Philippines, Diliman, Quezon City 1101, Philippines
\* Correspondence: glyn.barrett@reading.ac.uk

**Abstract:** We compared the covering behavior of four sea urchin species, *Tripneustes gratilla*, *Pseudoboletia maculata*, *Toxopneustes pileolus*, and *Salmacis sphaeroides* found in the waters of Malapascua Island, Cebu Province and Bolinao, Panagsinan Province, Philippines. Specifically, we measured the amount and type of covering material on each sea urchin, and in several cases, the recovery of debris material after stripping the animal of its cover. We found that *Tripneustes gratilla* and *Salmacis sphaeroides* have a higher affinity for plant material, especially seagrass, compared to *Pseudoboletia maculata* and *Toxopneustes pileolus*, which prefer to cover themselves with coral rubble and other calcified material. Only in *Toxopneustes pileolus* did we find a significant corresponding depth-dependent decrease in total cover area, confirming previous work that covering behavior serves as a protection mechanism against UV radiation. We found no dependence of particle size on either species or size of sea urchin, but we observed that larger sea urchins generally carried more and heavier debris. We observed a transport mechanism of debris onto the echinoid body surface utilizing a combination of tube feet and spines. We compare our results to previous studies, comment on the phylogeny of sea urchin covering behavior, and discuss the interpretation of this behavior as animal tool use.

**Keywords:** sea urchins; echinoid; echinoderms; covering behavior; animal tool use

## 1. Introduction

Several species of sea urchins cover their exposed body surfaces, in a special form of crypsis, with debris collected from their environment [1–5]. The debris can be biotic or abiotic in nature, and include scleractinian coral rubble, mollusc shells, sea grass, terrestrial leaves, and human rubbish. The sea urchins actively manipulate these debris fragments with their tube feet and combine these with their spines to transport them onto their upward facing body surface. Suggested reasons for this behavior are UV protection (demonstrated for *Tripneustes gratilla* [5,6]), olfactory camouflage (suggested for covering species living in light-deficient deep-sea environments [7,8], and use as a ballast for weighing down of the animal by increasing the relative density of the sea urchin in the face of currents or surge (shown for *Strongylocentrotus droebachiensis* [2] and for *Paracentrotus lividus* [9]) as well as a direct anti-predatory function [1,10]. It has been shown that sea urchins increase their coverage if faced with stressors such as UV radiation and wave action [2,6]. Urchins were even capable of distinguishing between two types of artificial covering material, and preferred the type that was more conductive to UV protection [5]. Furthermore, since the covering material often acts as an effective platform for algal growth and is often moved from and across the main body of the animal

toward the mouth, situated on the aboral surface, covering has been suggested as a food storage behavior [11,12].

Different species of sea urchins have been shown to be highly selective for the covering material used and make distinct choices for this [13]. For example, *Toxopneustes pileolus* showed a preference for the largest pieces of debris available amongst a range of differently sized substratum [14]. In tank studies, and given a choice of items of different sizes, *Lytechinus variegatus* selected those of intermediate size [15] in what the authors suggest is a trade-off between physiological energy expenditure and levels of UV protection. Individual choice of covering material in *Paracentrotus lividus* was found to be dependent on the size of the animal together with the availability of substrate [16], suggesting that covering behavior may differ dependent on habitat.

Whilst there are many studies on choice in the covering behavior of sea urchins, both at a species and individual level, there are few reports in the literature focusing on this behavior in a wider, ecological context whilst simultaneously considering multiple species sharing a common habitat. The aim of this study was to compare the covering behavior of four species of sea urchin, *Tripneustes gratilla*, *Pseudoboletia maculata*, *Salmacis sphaeroides*, and *Toxopneustes pileolus*, which are species found near reefs at shallow depths in the tropical Indo-Pacific, and specifically the Philippines. These species partially overlap in their habitat use, and were often seen alongside each other; hence, they have access to the same types of covering material and forage that is available in similar concentrations and ratios throughout the range. This makes differences in their covering behavior all together intriguing. We combine field observations of debris amount and type with video recordings of coverage material handling both in the field and in aquaria.

Given the active choice of coverage material and the situation-dependent loading of material, as well as the coordinated lifting of coverage material, we propose in the discussion that this behavior constitutes a rudimentary form of animal tool use.

## 2. Materials and Methods

Echinoids were sampled on self-contained underwater breathing apparatus (SCUBA) at depths of between 2–16 m around Malapascua Island, Cebu Province, and Bolinao, Pangasinan Province, Philippines. All four investigated species are moderately common at and around the sampling sites. Of the six sites we sampled, one site was populated by four, three, and one species of the collector sea urchins we investigated, respectively, and the three remainder sites (including the sampling site in Bolinao) was populated by two species. In the field, each sea urchin was photographed, measured (radial diameter), and carefully stripped of its attached debris. The debris was placed in a marked, plastic, Ziploc® bag and transported back to land via the research vessel. The "naked" sea urchin was then observed, and in many cases, filmed for up to 10 minutes, recording its covering behavior. If a sea urchin failed to re-cover itself, we would place debris comparable to the amount of debris removed from the animal to avoid any undue stress to the sea urchin, before moving to the next specimen. Post-dive, collected debris fragments were laid onto white slates with a ruler for scale and photographed directly from above. Individual pieces were counted, categorized, and surface area was determined using ImageJ® [17] image analysis software (Figure 1). Fragments were subdivided into the following categories: coral rubble, mollusc shells, tunicates, marine plants and algae, land plants, and human refuse. Fragments of debris were sun-dried for a period of 24 h and subsequently weighed using an electronic scale.

To determine the speed of re-covering, we traced the position of selected pieces of debris during the sea urchin's efforts to cover its body surface with debris. Footage was either shot in the field (by a diver with a hand-held camera) or in concrete aquarium tanks (with a camera positioned above the tank). Video analysis was conducted with Tracker 5.0 Video Analysis and Modeling Tool software (GNU General Public License, ©2017 Douglas Brown). At intervals of 10 s, the position of the sea urchin's pole and the pole-most point of the fragment were manually indicated, and the resulting positions were used for the analysis of fragment movement speed. At each time step, the scale bar

was also adjusted if necessary (if the distance from the camera to the sea urchin slightly changed in the case of hand-held footage). The radial position of the fragments determined in this way is an underestimation of the actual distance traveled, since the sea urchin bodies are compressed spheroids. However, relative measured speeds and positions are useful for comparative purposes.

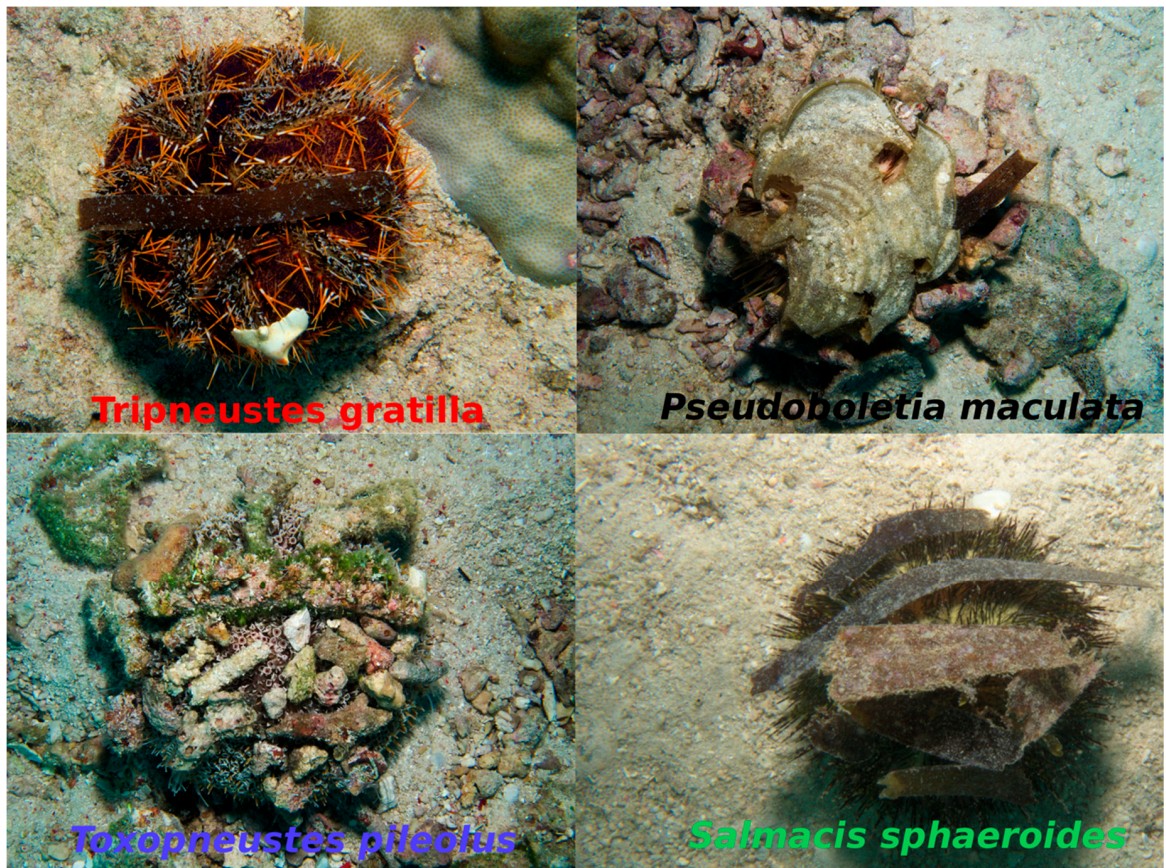

**Figure 1.** Examples of sea urchins covered in debris, photographed in their natural environment: *Toxopneustes pileolus, Tripneustes gratilla, Salmacis sphaeroides,* and *Pseudoboletia maculata.*

Six sea urchins each of the species *Tripneustes gratilla* and *Salmacis sphaeroides* were collected in Bolinao and maintained in aquaria at the Marine Lab of the University of the Philippines. The sea urchins were fed with seagrass and provided field-collected seagrass and coral rubble as coverage material. Close-up views of the covering behavior were filmed in the tanks.

Some of the video footage analyzed for this study can be accessed on our dedicated YouTube.com site as defined in Supplementary Materials.

*Statistical Analysis*

Levene's test was used to assess the homogeneity of variance between individual sea urchin species. We used ANOVA to compare debris type preference between species of collector sea urchins and the size of the sea urchins, as determined by radial diameter and individual and global fragment size and weight. Linear regression was used to predict the depth-dependent total cover area per sea urchin species and test for significance in the size and weight of fragments on the overall size of the sea urchin.

## 3. Results

We sampled a total of 49 sea urchins, *T. gratilla* (n = 11), *P. maculata* (n = 10), *S. sphaeroides* (n = 12), and *T. pileolus* (n = 16) across six sites. We compared debris type preference between species of

collector sea urchins, and found significant differences in coverage material most importantly between total organic (e.g., seagrass, rhodoliths, tunicates) (ANOVA, $F_{(3, 45)}$ = 7.198, $p$ < 0.001) and total inorganic (e.g., coral, shell, rock) (ANOVA, $F_{(3, 45)}$ = 8.654, $p$ < 0.001) materials (Figure 2). Post-hoc Tukey's Honest Significant Difference (HSD) analysis revealed that *S. sphaeroides* preferred seagrass as a covering material, whilst *T. pileolus* preferred coral fragments.

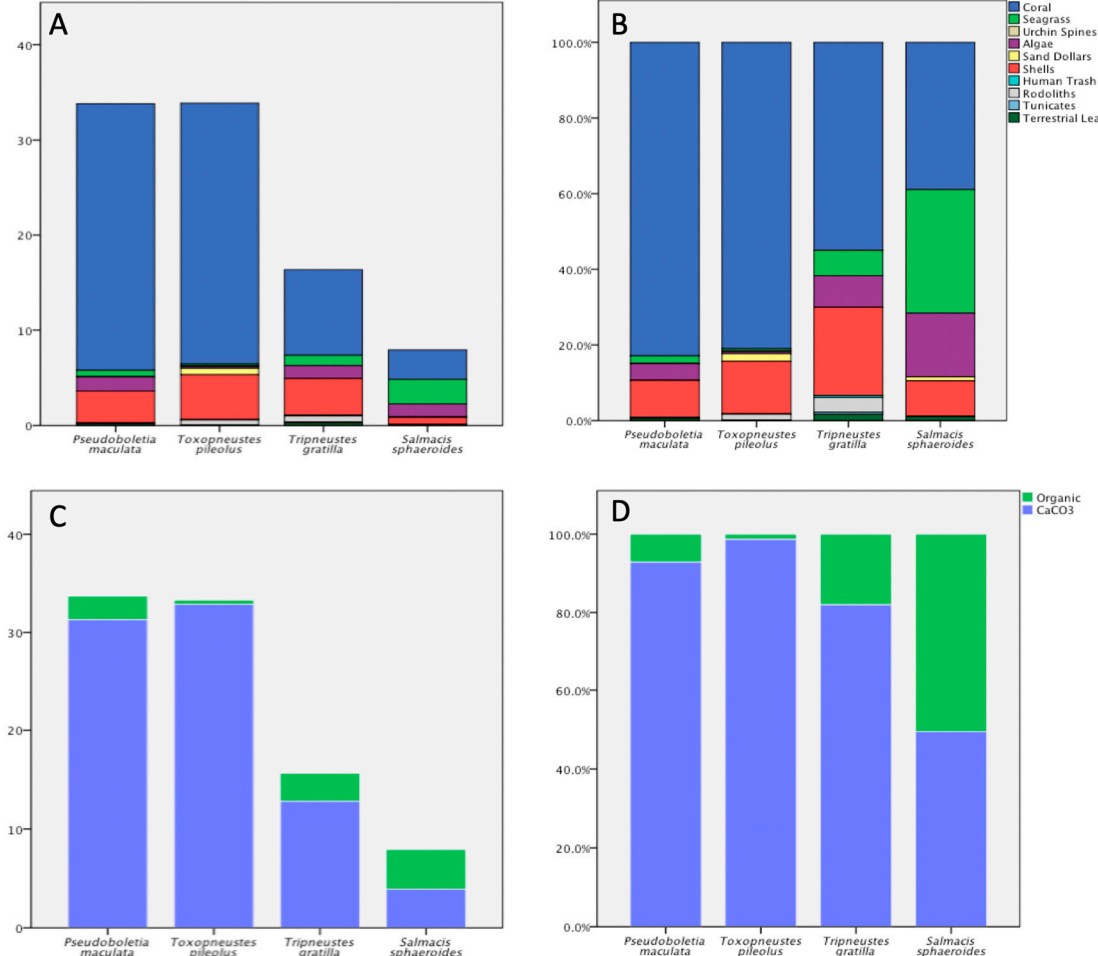

**Figure 2.** Sea urchin species-dependent preference in cover material. (**A**) Absolute numbers of debris type per species; (**B**) Relative contribution in % of each type of debris per species; (**C**) Absolute numbers of fragments per species sorted between calcified CaCO$_3$ skeletons and organic material; (**D**) Relative contribution in % of CaCO$_3$ skeletons and organic materials per species.

Linear regression was calculated to predict the total cover area (cm$^2$) based on depth (m) (Figure 3). A significant regression equation was found for *T. pileolus* ($F_{(1, 14)}$ = 7.443, $p$ = 0.016), with an R$^2$ of 0.347. *T. pileolus* predicted that the total cover area is equal to 248.4 – 10.1 ($x$) cm$^2$ where ($x$) is depth measured in m. Therefore, sea urchin cover area decreased by an average of 10.1 cm$^2$ for each meter of depth. Interestingly, depth was not a significant factor in influencing the total cover area for *T. gratilla*, *P. maculata*, or *S. sphaeroides*.

We found significant differences in size, as defined by equatorial diameter, of species of sea urchin (ANOVA, $F_{(3, 45)}$ = 211.722, $p$ < 0.001) with *S. sphaeroides*, *P. maculata*, *T. gratilla*, and *T. pileolus* at 6.3 cm, 9.6 cm, 9.7 cm, and 14.8 cm, respectively (Figure 4). Interestingly, the global mean size of fragments (5.6 ± 0.68 cm$^2$) was independent of sea urchin species and the size of the sea urchin (ANOVA, $F_{(1, 48)}$ = 0.7, $p$ = 0.407) (Figure 5). Linear regression was calculated to predict both the total number ($n$) and total weight (g) of fragments based on the diameter of the sea urchin (cm). Significant regression equations were found for both the total fragment number ($F_{(1, 47)}$ = 10.094, $p$ = 0.003),

with an $R^2$ of 0.177, and the total weight of fragments ($F(1, 47) = 3.997$, $p = 0.05$), with an $R^2$ of 0.078. The predicted total fragment number is equal to 8.818 + 0.071 (diameter) n when the diameter is measured in cm. The predicted total fragment weight is equal to 9.676 + 0.011 (diameter) g when the diameter is measured in cm. The total fragment number and total fragment weight increased, respectively, by 0.071 g and 0.011 g for each increase of 1 cm in diameter.

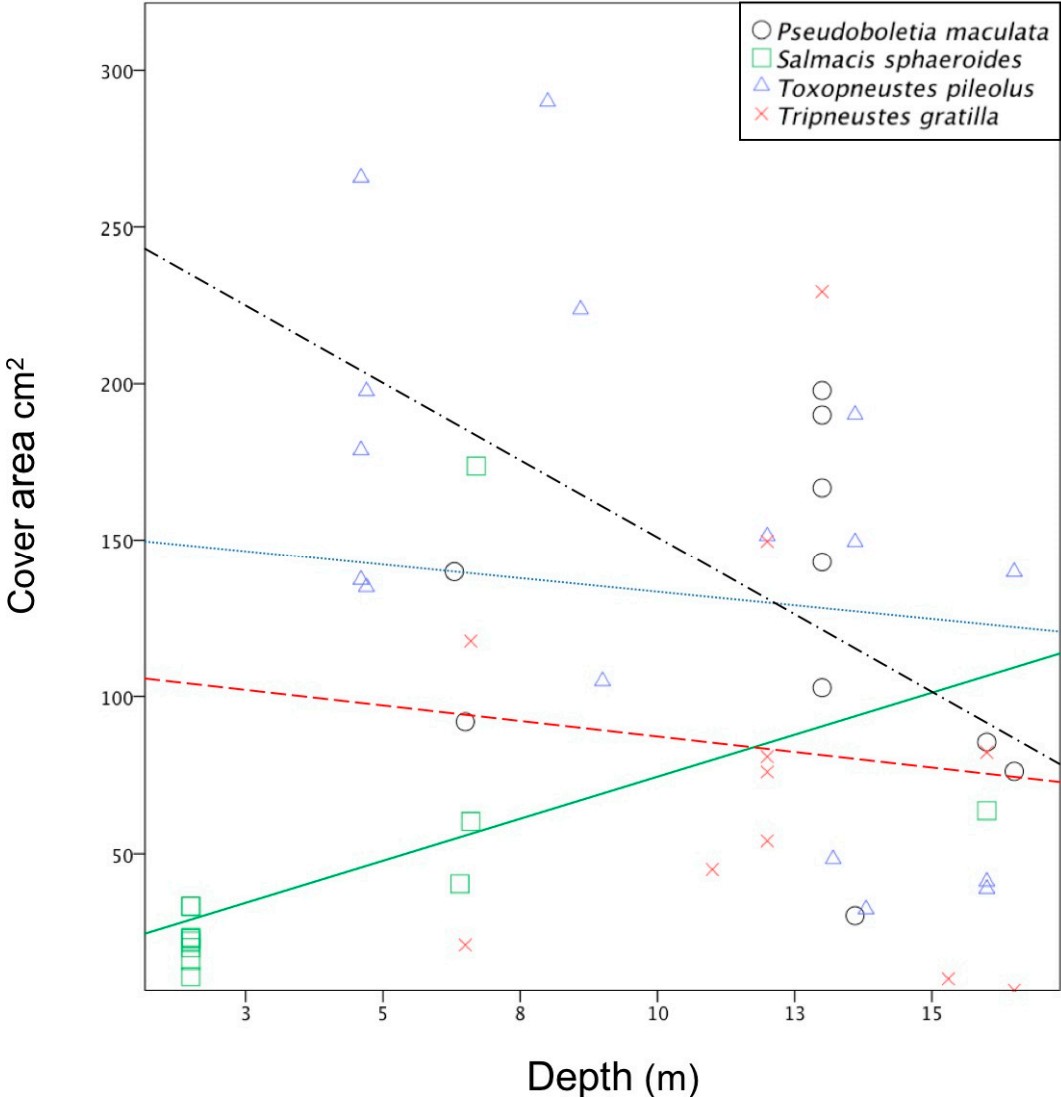

**Figure 3.** Depth influences the amount of debris on collector sea urchins. Fragment area for *Tripneustes gratilla* (red crosses), *Pseudoboletia maculata* (black circles), *Salmacis sphaeroides* (green squares), and *Toxopneustes pileolus* (blue triangles).

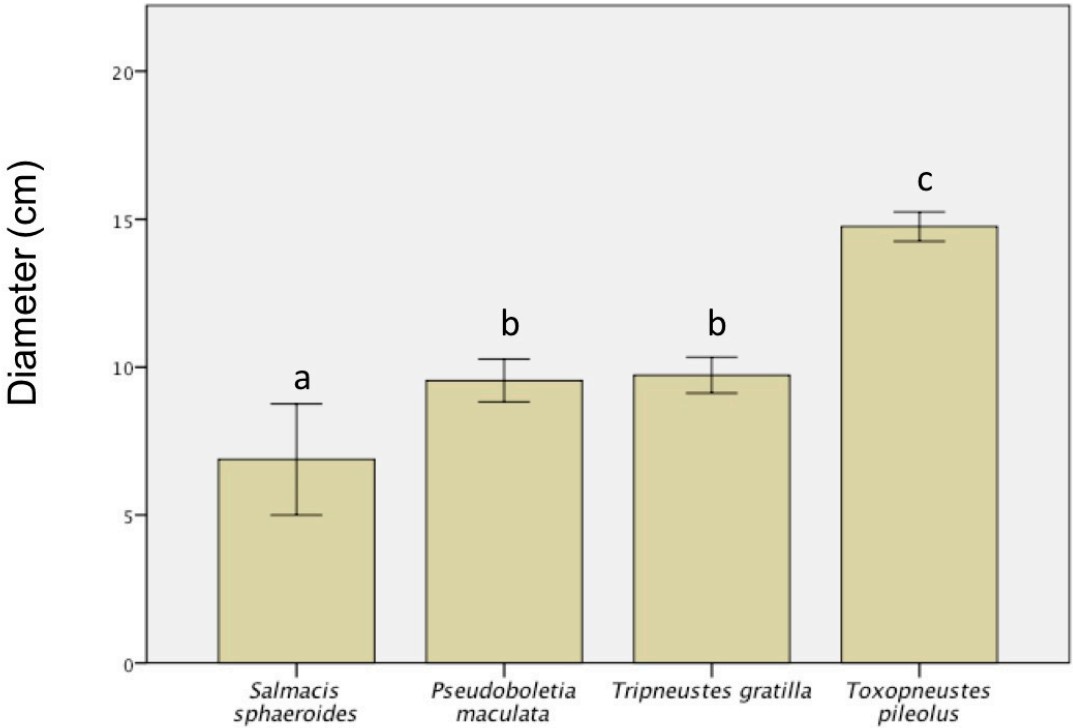

**Figure 4.** Diameter (mean ± SEM) of collector sea urchins. Means with different letters are significantly different (Tukey's HSD, $p < 0.05$).

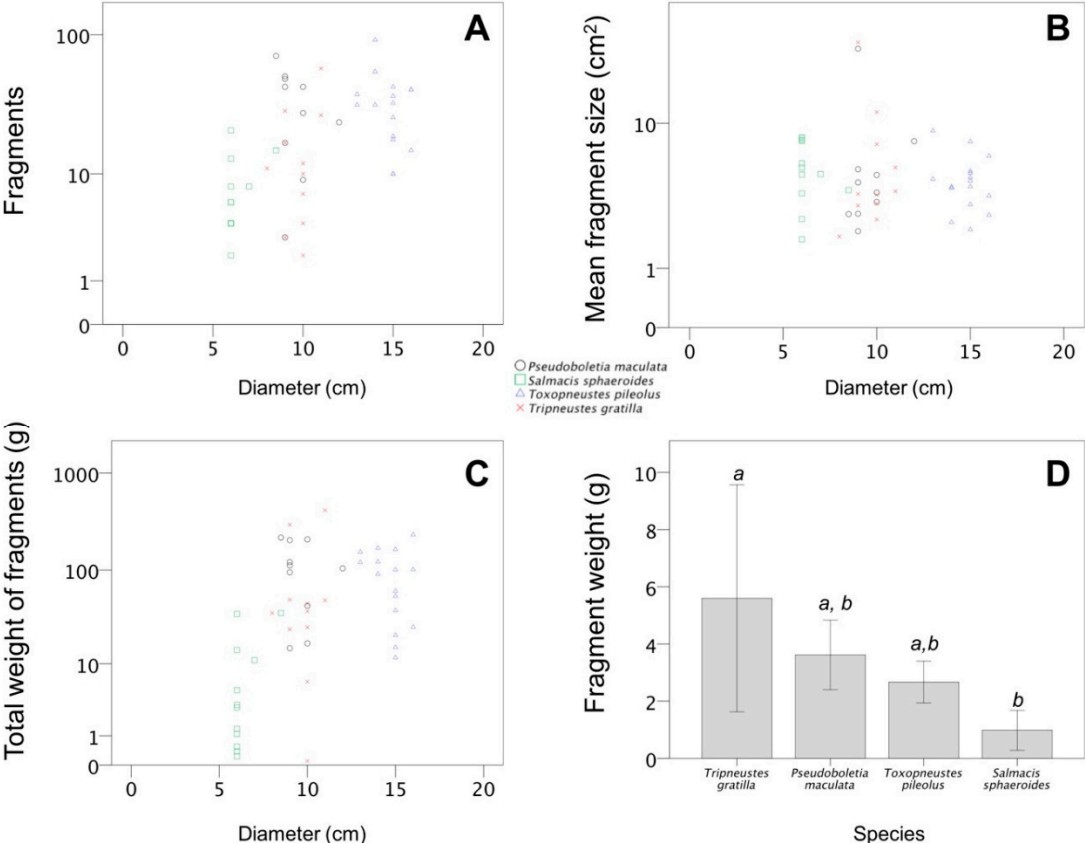

**Figure 5.** (**A**) Diameter vs. total fragments; (**B**) Diameter vs. average size of fragment (cm²); (**C**) Diameter vs. total weight of fragments (g); *x*-axis is diameter (cm); *y*-axis is $\log_{10}$ scale. (**D**) Average fragment weight per species.

Mean fragment weight (±SEM) in g per species was as follows: *Pseudoboletia maculata*: 3.62 ± 0.54, *Toxopneustes pileolus*: 2.67 ± 0.34, *Tripneustes gratilla*: 5.6 ± 1.78, and *Salmacis sphaeroides*: 0.98 ± 0.32. An analysis of variance (ANOVA) of the observed fragment weights yielded significant variation among sea urchin groups, $F(3, 45) = 4.615$, $p = 0.007$. A post-hoc Tukey test, denoted by letters above error bars, showed significance among groups at $p < 0.05$. The mean difference in the fragment weight of *S. sphaeroides* and *T. gratilla* was significant at the value of $p = 0.004$.

After we removed the debris coverage from the body of a sea urchin, it would quickly grab new debris with its tube feet and place it on top of its body. The speed of fragment movement onto the sea urchin was similar between different species (Figure 6A). In the footage we analyzed, *P. maculata* moved coral rubble the fastest, at a mean distance of 2.0 cm min$^{-1}$, followed by *T. pileolus* at 1.91 cm min$^{-1}$, and *T. gratilla* at 0.99 cm min$^{-1}$. The transport speed was not constant during the movement of a fragment, often with periods of fast transport being preceded and followed by period of slower movement (Figure 6B). Hence, despite the divergent global speeds noted above, instantaneous transport speed (as seen in the slopes of the fragment positions in Figure 6B) was relatively similar. Movements in the efforts of orienting individual fragments also affected the speed of its movement toward the dorsal pole of the echinoid.

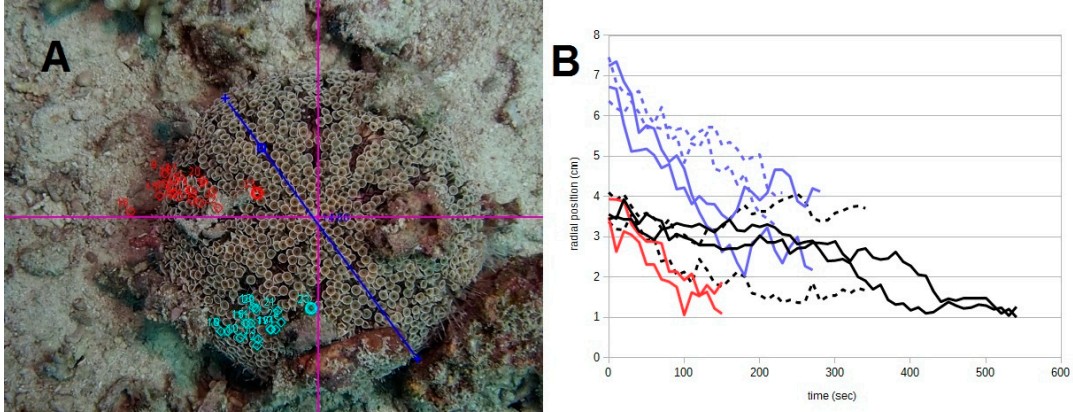

**Figure 6.** Speed of recovery of debris cover of sea urchins. (**A**) Points indicating movement trajectory of the pole-most point of a coral rubble fragment on *Toxopneustes pileolus* (in the field). Note the scale bar in blue. Screen-shot from Tracker software; (**B**) Position of fragment relative to the dorsal pole of the sea urchin. Species are color coded (*P. maculata*, red; *T. pileolus*, blue; *T. gratilla*, black), continuous and broken lines indicate fragments on different animals. The footage was recorded in tanks (*T. gratilla* solid line, both *S. sphaeroides*) and in the field (all other recordings).

The covering behavior is jointly carried out by spines and tube feet. Initially, the tube feet make searching rotational movements, until one of them comes into contact with a piece of debris. The tube foot is soon joined by several other tube feet, which jointly pull the fragment toward the sea urchin's body (Figure 7A,B). It is not obvious from the video observations whether this is due to a directed recruitment (i.e., concerted effort) of tube feet by the echinoid nervous system, or simply due to the proximity of neighboring tube feet to the detected fragment. Next, the tube feet pull the debris fragment closer to the echinoid. The fragment is most commonly transported up onto the body of the sea urchin by a combined pulling movement of the tube feet and a pushing movement of the spines. Upward-angled spines keep the fragment from slipping off, while the spines above the fragment flatten out to allow fragments to pass over freely. The tube feet above the fragment are often seen pulling it toward the pole (anus) of the echinoid (Figure 7C,D). The pedicellariae are not involved in the transport process. The fragments reaching the top of the sea urchin naturally block the transport of further lower fragments, which is a process that repeats until the complete dorsal surface of the sea urchin is covered. We often observed that the sea urchins moved in one direction when stripped of

covering material, whilst re-covering. This combined covering-walking behavior was observed both in the field and in tanks.

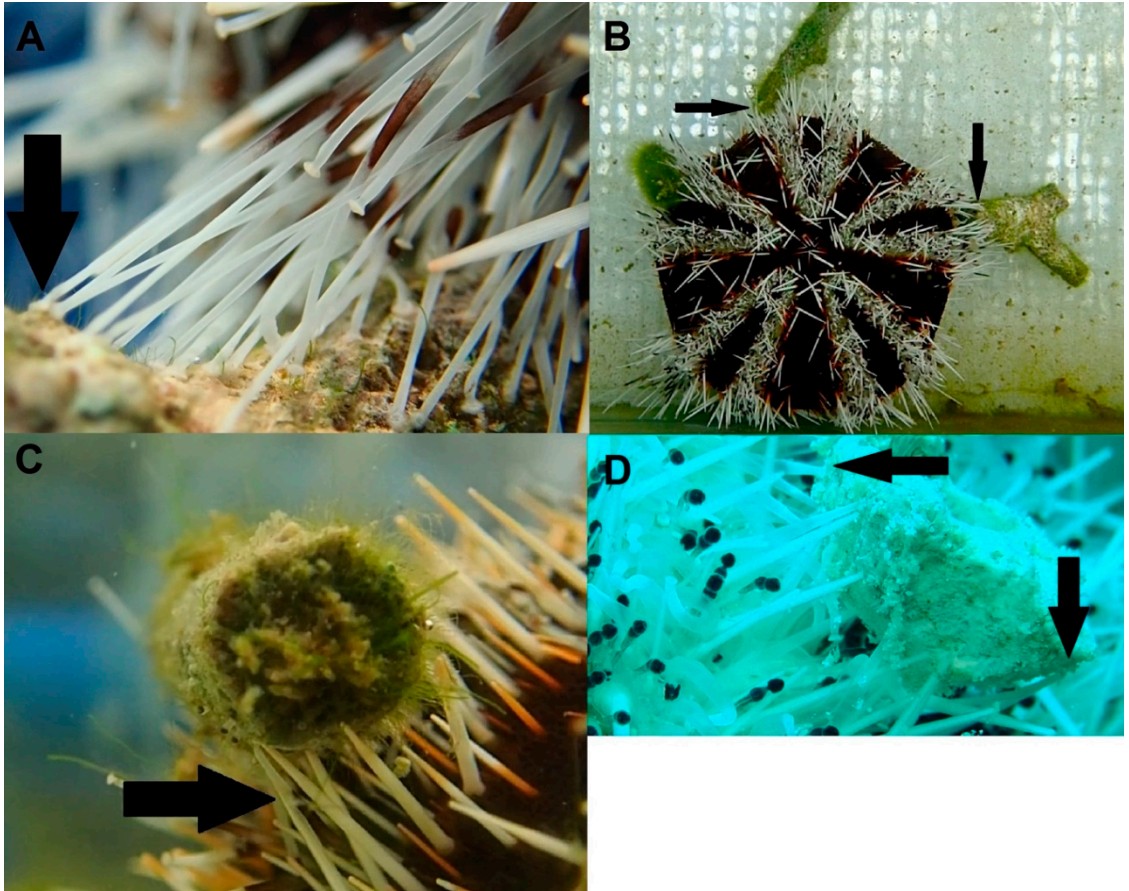

**Figure 7.** Spines and tube-feet are involved in the covering behavior in *Tripneustes gratilla*. Screen-shots of videos. (**A**) Several tube feet (one indicated by an arrow) jointly grab a coral rubble fragment in the initial phase of transport; (**B**) Two fragments held by tube feet (arrows); (**C**) Spines pushing a fragment upwards; (**D**) Spines pushing (downward arrow) and tube feet pulling (leftward arrow) a fragment. (**A**,**D**) are from footage recorded in the field; (**B**,**C**) were recorded in tanks.

Several anecdotal observations during our study are worth mentioning. Two of the sea urchins carried seemingly living tunicates on their backs when first encountered. We found many sea urchins in areas with sand or coral rubble that was interspersed by rocks not higher than 30 cm. Tunicates sit on the edges of many of these rocks, exactly in a position where a passing sea urchin would be in the position to impale them. Another observation we repeatedly made was that sea urchins stripped of debris would crawl underneath the protective spines of groups of nearby *Diadema* sp. sea urchins. Also, when the collector sea urchins replaced the debris covering their bodies after we removed it, they caused significant perturbation of the sand and rubble they "walked" over as if they were attempting to dig themselves into the substrate. Then, the sea urchins were often followed by several species of wrasse (mostly *Choris batuensis*, *Oxychelinus* sp., *Diproctacanthus* sp.) and sand perch (*Parapercis* sp.), presumably aiming to acquire food items from amongst the disturbed substrate.

## 4. Discussion

Our study shows that the covering material of four species of collector sea urchin, *Pseudoboletia maculata*, *Toxopneustes pileolus*, *Salmacis sphaeroides*, and *Tripneustes gratilla,* is distinctively species-dependent. A previous study of the covering behavior of *Tripneustes ventricosus* and *Lytechinus variegatus* in the Caribbean found that the former prefers to cover itself with seagrass, even though it

uses other material independently of seagrass availability [13]. Our study comes to a similar conclusion, with 15% of the cover of the closely related *Tripneustes gratilla* being seagrass, compared to 0.5% and no seagrass cover for *Toxopneustes pileolus* and *Pseudoboletia maculata,* respectively. *Salmacis sphaeroides* had a higher preference for organic material with a total of 52%; 87% of this material was seagrass alone. Hence, our study confirms a species-specific choice in covering material. That the fragment size was similar for different sea urchin species of different sizes points to similar covering mechanisms in all these species. A video analysis of fragment transport agrees with this conclusion, and shows a covering mechanism involving both spines and tube-feet.

We only found a depth-dependent decrease in debris coverage in *Toxopneustes pileolus,* which confirms the purpose of the covering behavior, in this species, as UV protection and a weighting down to counter water movement, as seen in previous studies [2,5,6,18]. The limited depth range at which we sampled the other species might have contributed to the lack of depth-dependence evidence in coverage.

### 4.1. Echinoid Covering Behavior Evolution

Phylogenetically, covering behavior occurs widely in Echinoidea. The species compared here belong to the families of the Toxopneustidae (*Pseudoboletia maculata* and *Toxopneustes pileolus),* and Temnopleuroidae *(Tripneustes gratilla* and *Salmacis sphaeroides).* Other echinoid families with members showing covering behavior are listed in Table 1. These families span several groups of echinoids outside of the more primitive pencil sea urchins (cidaroids) and the long-spined *Diadema* species (Figure 7). Among the Regularia (radially symmetrical sea urchins), covering behavior is found in several groups of the Echinacea. A big cluster of collector sea urchins is found in the Camarodonta, including collectors in the families of the Parechinidae, Strongylocentrotidae, Temnopleuridae, and Toxopneustidae. We found two mentions of irregular echinoids (asymmetric sea urchins, sand dollars, and their relatives) with covering behavior. One covering sea urchin is the deep-sea urchin *Conolampas sigsbei,* belonging to family Echinolampadidae in the Neognathostomata branch of the irregularia [3,19]. Also, *Cystochinus loveni,* another deep-sea species, was observed to cover itself with a layer of protists. The authors noted that the covering behavior was most likely specifically loaded by the echinoid, and speculated that camouflage and a weighing-down effect are the reasons for this covering behavior [20]. We are aware that many Irregularia bury themselves in the sediment [21]. This behavior, which likely shares mechanistic features with the covering behavior discussed here, is nevertheless distinct in its ecological role, and we treat the two behaviors separately. The ancestors of these collector sea urchin families split around 200 Ma ago in the lower Jurassic [22]. Since collecting behavior has not been observed in the majority of echinoids, it presumably evolved convergently multiple times in the lineages mentioned here (for an extensive discussion of convergent evolution, see [23]).

**Table 1.** A list of echinoids showing covering behavior, assembled from the literature and from personal observations. We make no claim that this list is exhaustive. See also Figure 7.

| Order | Family | Genus | Species | Distribution | Reference |
|---|---|---|---|---|---|
| Camarodonta | Parechinidae | *Paracentrotus* | *lividus* | Mediterranean Sea and eastern Atlantic Ocean | [16] |
| Camarodonta | Temnopleuridae | *Salmacis* | *sphaeroides* | Tropical Indo-Pacific | [24] |
| Camarodonta | Toxopneustidae | *Lytechinus* | *anamesus* | Caribbean | [25] |
| Camarodonta | Toxopneustidae | *Lytechinus* | *euerces* | Deep sea, Caribbean | [26] |
| Camarodonta | Toxopneustidae | *Lytechinus* | *variegatus* | Caribbean | [13] |
| Camarodonta | Toxopneustidae | *Toxopneustes* | *roseus* | Gulf of California | [27] |
| Camarodonta | Toxopneustidae | *Tripneustes* | *gratilla* | Tropical Indo-Pacific | This study |
| Camarodonta | Toxopneustidae | *Tripneustes* | *ventricosus* | Caribbean | [13] |
| Camarodonta | Toxopneustidae | *Pseudoboletia* | *maculata* | Tropical Indo-Pacific | This study |
| Camarodonta | Toxopneustidae | *Toxopneustes* | *pileolus* | Tropical Indo-Pacific | This study |
| Echinoida | Strongylocentrotidae | *Strongylocentrotus* | *droebachiensis* | Temperate Atlantic | [18] |
| Echinoida | Strongylocentrotidae | *Strongylocentrotus* | *intermedius* | Temperate and cold northern hemisphere | [1] |
| Cassiduloida | Echinolampadidae | *Conolampas* | *sigsbei* | Deep sea, Caribbean | [26] |
| Spatangoida | - | *Paleobrissus* | *hilgardi* | Deep sea, Caribbean | [26] |
| Stomopneustoida | Glyptocidaridae | *Glyptocidaris* | *Crenularis* | Temperate and cold northern hemisphere | [3] |

### 4.2. Echinoid Covering Behavior as Tool Use

An intriguing question is if the covering behavior shown by sea urchins can be classified as animal tool use. Animal tool use is notoriously difficult to define, especially in borderline cases [28]. Following previous work on the matter of tool use in finches [29] and primates [30], St Amant and Horton in 2008 [31] expanded the definition by Shumaker et al. [32] and defined tool use as " . . . *the exertion of control over a freely manipulable external object (the tool) with the goal of: (1.) altering the physical properties of another object, substance, surface or medium (the target, which may be the tool user or another organism)* via *a dynamic mechanical interaction; or (2.) mediating the flow of information between the tool user and the environment or other organisms in the environment*". Collector sea urchins manipulate and freely control marine debris fragments of their choice with the goal of altering the physical properties of themselves (the tool user), for the sake of UV, predation, and water surge protection. They also mediate the flow of information between the tool user and the environment or other organisms, by using the debris as camouflage.

Thus, the collecting behavior of sea urchins falls into two categories, as given in Bentley-Condit and Smith (2010), namely, physical maintenance (i.e., the use of a tool to affect one's appearance or body) and predator defense (i.e., the use of a tool to defend oneself). Bentley-Condit and Smith (2010) cited the placement of anemones on their shells by hermit crabs as an example of the latter, which is quite similar to the echinoderm behavior discussed here. Lawrence [33] argued that the covering behavior of *Lytechinus variegatus* is reflexive, rather than functional. This, in our opinion, lumps proximate and ultimate causes [34,35]. Certainly the proximal cause, which is also the physiological and behavioral explanation, is a reflexive loading of debris by the sea urchin. The animals achieve this reflexive loading without any form of understanding of their actions (see the following paragraph). However, the ultimate cause of the debris covering, the adaptive value of the behavior formed in evolution, is the use of debris as a tool, as indicated by the selectivity of debris collected and the modification of debris cover in response to different environmental situations.

A significant amount of investigation and experimentation has been directed to the question of how much reasoning and insight is behind animal tool use. Tool use by humans, and presumably apes and crows, derives from a mental model of the physical effects of the tool (for a good review, see [36]). However, such an internal model *is not a condition* for tool use, and Seed and Byrne (2010) reasoned that insect tool use (for example, ant lions flicking sand at prey) comes about without an internal representation of the tool as an extension of the animal's body. Collector sea urchins most likely have no mental representation of the tools they are using or the intended consequences of the tool use, since they lack a centralized brain. Nevertheless, several observations indicate that echinoid covering behavior is tool use, as defined in the literature cited above:

- The use of *selected* pieces of debris as a cover by sea urchins (as shown in this and other studies [5,13]). By no means is the covering material simply debris that gets stuck on the sea urchin, but it is there through the active selection and choice of the animal, as a consequence of the coordinated action of tube-feet and spines in response to the environment.
- The *adjustment of debris cover according to need* (increased coverage in response to increased UV radiation [2], and the increased coverage at shallower depth as demonstrated in this study) also point to goal-directed tool-use behavior.
- The *difference in debris type between echinoid species* points toward a specific form of tool use tailored to the needs and capabilities of the different sea urchin species.

We are by no means the first investigators to observe and describe the covering behavior of sea urchins, but the conceptual connection between this behavior and tool use has, to our knowledge, not been made previously, besides a passing mention in one other study [37]. We do believe that this behavior can be classified as tool use, adding the Echinodermata as a fourth phylum with members capable of tool use next to the Chordata (mainly primate mammals and corvid Aves), Arthropoda, and Mollusca (primarily octopod cephalopods) [28]. Alternatively, the commonly used definitions of

tool use will have to be narrowed to exclude tool use without an explicit mental representation of the tool and its intended effects.

**Supplementary Materials:** A video showcasing sea urchin covering behaviour through the use of spines and tube feet is available on our Youtube site (https://www.youtube.com/watch?v=GZw0mKy_un8&t=4).

**Author Contributions:** All of the authors collected data; G.A.B. conceived the idea; G.A.B. and K.M.S. analyzed the data and wrote the manuscript.

**Funding:** This research received no external funding.

**Acknowledgments:** We thank everyone at People and the Sea and at the UP Bolinao Marine Lab for their support, as well as Cecilia Conaco and Patrick Cabaitan for their helpful discussion of the manuscript.

**Conflicts of Interest:** The authors declare no conflict of interest.

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
