# Peer review of "Tool Use by Four Species of Indo-Pacific Sea Urchins"

_jmse, doi:10.3390/jmse7030069_

Round 1

Reviewer 1 Report

Original primary research that falls within the scope of the journal. The Research question are not clearly defined, though relevant

& meaningful. Also the introduction needs more details, in order to

provide more justification for the study (in particular, it should be expanded on the basis of the gap of knowledge that is filled).

Methods are described with sufficient detail &

information to replicate.

Conclusions are well stated, linked to

original research.

Author Response

·       Original primary research that falls within the scope of the journal. The Research question are not clearly defined, though relevant & meaningful.

Our response – Thank you. The research question has been elaborated upon in the main body of the text

·       Also the introduction needs more details, in order to provide more justification for the study (in particular, it should be expanded on the basis of the gap of knowledge that is filled).

Our response – Further details have now been added to the introduction.

·       Methods are described with sufficient detail & information to replicate. Conclusions are well stated, linked to original research.

Our response – Thank you!

Reviewer 2 Report

This paper needs to have numerous corrections - there are careless misspellings of
scientific words; several words are not capitalized.

Review of previous studies of covering behavior is superficial and very
incomplete, and it had led to misinterpretations of the work of previous
authors. Note, for example, that, contrary to what they state, covering
has been reported in more than one species of "irregular" echinoids.

The discussion of "tool use" is unconvincing to this reviewer. The
authors do not take into account the notion that accumulation of debris
by "covering urchins" may include a strong element of reflexive
acquisition of debris - as discussed by Lawrence (1976) and others.  This neglect of the work of others potentially leads to a hasty misinterpretation of the urchin's actions!

Author Response

·      This paper needs to have numerous corrections - there are careless misspellings of
scientific words; several words are not capitalized.

Our response - Thank you. These have now all been amended in the text. References to phylum, super-order and order have now been capitalised.

·      Review of previous studies of covering behavior is superficial and very
incomplete, and it had led to misinterpretations of the work of previous
authors. Note, for example, that, contrary to what they state, covering
has been reported in more than one species of "irregular" echinoids.

Our response - Thanks for the suggestion. The introduction has now been improved and a further reference on the irregularia added.

·      The discussion of "tool use" is unconvincing to this reviewer. The authors do not take into account the notion that accumulation of debris by "covering urchins" may include a strong element of reflexive acquisition of debris - as discussed by Lawrence (1976) and others. This neglect of the work of others potentially leads to a hasty misinterpretation of the urchin's actions!

Our response – As suggested by the reviewer, we have added the following reference :

Ferber, I., & Lawrence, J. M. (1976). Distribution, substratum preference and burrowing behaviour of Lovenia elongata (Gray)(Echinoidea: Spatangoida) in the Gulf of Elat ('Aqaba), Red Sea. Journal of Experimental Marine Biology and Ecology22(3), 207-225.

Regarding tool use, we have added the following to main body of the text

“Alternatively, the commonly used definitions of tool use will have to be narrowed, to exclude tool use without an explicit mental representation of the tool and its intended effects.”

Round 2

Reviewer 2 Report

On line 261 cidaroids is misspelled.

In Table 1, Paleobrissus is misspelled, and it does NOT belong in the Toxopneustidae; it is an "irregular" echinoid in the Order Spatangoida. 

Levin et al. (2001) document covering in yet another irregular echinoid - Cystechinus loveni (JMBA UK 81:881-882).

Lawrence (1976; Nature 262:491-2) is of particular importance to your arguments, and yet you still do not mention this publication..this makes me wonder why???

The mistakes, and the seemingly careless omission of some highly relevant and important publications in their manuscript, will result in much criticism if this paper is published in its present form.

Author Response

Reviewer Comment: On line 261 cidaroids is misspelled.

Our response: corrected in text

Reviewer Comment: In Table 1, Paleobrissus is misspelled, and it does NOT belong in the Toxopneustidae; it is an "irregular" echinoid in the Order Spatangoida.

Our response: corrected in text. Order has now been added to table and the list categorised in alphabetical order

Reviewer Comment: Levin et al. (2001) document covering in yet another irregular echinoid - Cystechinus loveni (JMBA UK 81:881-882).

Our response: Absolutely valid comment thank you. The reference has now been added to the text with further explanation.

Reviewer Comment: Lawrence (1976; Nature 262:491-2) is of particular importance to your arguments, and yet you still do not mention this publication..this makes me wonder why???

Our response: We apologize for not having included Lawrence, Nature 1976, as advised initially. We had thought the reviewer had referred to another paper by Lawrence from that same year. We simply had missed the reference (papers of that age are not keyworded well, unfortunately), it has included now together with explanatory text. It actually raises a very important point, and we have added 2 more references to clarify this point. Thank you for pointing us at the reference!